# RETHINKING FAIR REPRESENTATION LEARNING FOR PERFORMANCE-SENSITIVE TASKS

**Charles Jones[1,†], Fabio De Sousa Ribeiro[1], Mélanie Roschewitz[1],**
**Daniel C. Castro[2] & Ben Glocker[1,†]**
[1]Department of Computing, Imperial College London, UK
[2]Microsoft Research Health Futures, Cambridge, UK
[†]Correspondence: {`charles.jones17,b.glocker`}`@imperial.ac.uk`

## ABSTRACT

We investigate the prominent class of fair representation learning methods for bias mitigation. Using causal reasoning to define and formalise different sources of dataset bias, we reveal important implicit assumptions inherent to these methods. We prove fundamental limitations on fair representation learning when evaluation data is drawn from the same distribution as training data and run experiments across a range of medical modalities to examine the performance of fair representation learning under distribution shifts. Our results explain apparent contradictions in the existing literature and reveal how rarely considered causal and statistical aspects of the underlying data affect the validity of fair representation learning. We raise doubts about current evaluation practices and the applicability of fair representation learning methods in performance-sensitive settings. We argue that fine-grained analysis of dataset biases should play a key role in the field moving forward.

## 1 INTRODUCTION

If we wish to deploy deep predictive models in high-stakes settings, such as medical diagnosis, we must understand and mitigate performance disparities across population subgroups (Buolamwini & Gebru, 2018; Seyyed-Kalantari et al., 2021). Despite considerable effort in developing methods for debiasing representations of deep models, little progress has been made towards understanding the validity of such methods for real-world deployment. Proposed methods often achieve state-of-the-art results on one benchmark, only to be beaten by conventional empirical risk minimisation (ERM; Vapnik, 1999) on more comprehensive evaluations (Zietlow et al., 2022; Zong et al., 2023). Further analyses have shown a concerning 'levelling down' effect (Mittelstadt et al., 2023), indicating that today's group fairness methods may even cause harm if deployed in the real world.

One aspect behind the apparent failure of fairness methods is an inconsistent approach to model evaluation. One prominent approach focuses on maximising subgroup performance for test data that are independent and identically distributed (IID) to training data, effectively ignoring dataset bias and treating fairness as a learning problem (e.g. Zietlow et al., 2022; Dutt et al., 2023). A second approach assumes that training data includes known spurious correlations and seeks to generalise to an out-of-distribution test set with the bias removed (e.g. Kim et al., 2019; Tartaglione et al., 2021). A third approach even ignores absolute performance entirely, aiming instead to enforce relative equality of properties such as predicted positive (Zemel et al., 2013) or true positive (Hardt et al., 2016) rates.

These three branches of research represent fundamentally different paradigms of fairness analysis; they make different ethical assumptions and require different methods, metrics, and benchmarks. Concerningly, however, much work leaves the distinction between these approaches implicit, and we often see methods from one paradigm employed (potentially inappropriately) in others. Specifically, we will consider the prominent class of *fair representation learning* methods (FRL; Zemel et al., 2013; Cerrato et al., 2024), which aim to remove sensitive information from learned representations. These methods were initially developed to enforce the demographic parity metric (i.e. enforcing an equal proportion of positive predictions in each group) but have since been applied in settings focusing on maximising IID performance (e.g. Pfohl et al., 2021; Zhang et al., 2022), or overcoming distribution shifts (e.g. Kim et al., 2019; Wang et al., 2020), with mixed results.

We apply tools from causal reasoning (Pearl, 2011) to clarify the distinctions between different paradigms in fairness analysis. We analyse implicit assumptions harming the validity of FRL methods when applied outside of the settings they were designed for, deriving theoretical results that explain apparent contradictions in the existing literature. Our results indicate that bias mitigation methods must be clearer about their assumptions and limitations, and we call on the community to be explicit about what problems the proposed methods aim to solve. Our contributions are:

§2 We provide a unifying perspective on the fairness literature by organising relevant work into three parallel streams, each representing different methodological and evaluation paradigms.

§3 We define causal structures representing realistic scenarios of dataset bias and discuss how the bias mechanisms may affect the performance and fairness of predictive models.

§4 We prove fundamental limitations on the validity of FRL methods when applied in IID settings and propose two hypotheses for the validity of FRL under distribution shift.

§5 We support our theoretical results and hypotheses with a comprehensive set of real-world experiments and discuss the implications of our results for the field moving forward.

## 2 THREE PARADIGMS OF GROUP FAIRNESS ANALYSIS

We begin by introducing three distinct paradigms of group fairness analysis from the literature, detailing how FRL methods have been applied in each one. Note that we do not make claims about the legitimacy or appropriateness of each paradigm – such decisions must be made with ethical knowledge of the application domain (Fazelpour et al., 2022; Mccradden et al., 2023). By organising the relevant literature into these three paradigms, we aim to clarify the consequences of (mis)applying FRL methods outside of the problems they were initially developed for.

**Enforcing group parity**  Some of the earliest and most influential research in fair machine learning focuses on enforcing equality of classifier properties across subgroups. This is the context in which Zemel et al. (2013) introduced FRL, a training strategy which prevents models from encoding sensitive information in their representations. In high-dimensional deep learning problems, FRL is typically implemented through either adversarial training (Edwards & Storkey, 2016; Alvi et al., 2018) or by applying disentanglement techniques (Creager et al., 2019; Sarhan et al., 2020). Variants of FRL have been applied in both supervised and unsupervised (Louizos et al., 2017) settings to enforce demographic parity on downstream predictive tasks (Madras et al., 2018). In the supervised case, FRL may be class-conditional (Zhao et al., 2020), corresponding to the equalised odds criterion (Hardt et al., 2016) instead. Beyond FRL, a large body of work in this paradigm focuses on understanding tradeoffs between group fairness metrics such as equal opportunity, calibration, and demographic parity (Kleinberg et al., 2016; Chouldechova, 2017; Kim et al., 2020; Friedler et al., 2021).

**Maximising (subgroup-wise) IID performance**  A notable aspect of the group parity paradigm is that equality is often achieved by worsening performance for some (or all) groups ('levelling down'; Wachter et al., 2021; Zietlow et al., 2022; Mittelstadt et al., 2023), which is likely unacceptable in performance-sensitive domains, such as medical diagnosis (Petersen et al., 2023; Weng et al., 2024). In such fields, we have seen a shift from considering fairness as a question of group parity to a goal of maximising performance for all groups (Martinez et al., 2020; Diana et al., 2021). Considerable effort has been made in applying FRL methods to this setting but with limited success. McNamara et al. (2019), Zhao & Gordon (2019), and Zhao et al. (2022) derive various negative theoretical results for the performance of FRL on IID tasks. While these results have been known for some time, there seems to remain confusion on this point in the literature, and we have seen repeated attempts to apply FRL in IID settings. Empirically, Pfohl et al. (2021), Zhang et al. (2022), Zietlow et al. (2022), and Zong et al. (2023) benchmark various FRL methods under IID assumptions, finding that they consistently underperform compared to ERM or alternative bias mitigation techniques.

**Generalising to unbiased distributions**  In the IID setting, any bias present in the training must also appear in the test set. Thus, maximising test-time performance may be undesirable, as it will likely encourage models to reflect whatever bias we were initially trying to remove. The third paradigm of research thus views fairness as a problem of generalising from a biased training dataset to an unbiased deployment setting (Kim et al., 2019; Wang et al., 2020; Tartaglione et al., 2021). In this context,

fairness and distribution shift are two sides of the same coin – a fair model, by definition, seeks to maximise subgroup-wise performance when generalising to an unbiased test set. This branch of work lends itself particularly well to causal analysis, which provides a unifying language for understanding shifts across groups and settings (Pearl & Bareinboim, 2011; Castro et al., 2020). Wachter et al. (2021), Anthis & Veitch (2024), and Jones et al. (2024) connect distribution shifts to assumed causal and ethical properties of the underlying data-generating process, relating causal notions of fairness (Kusner et al., 2017; Chiappa, 2019; Plečko & Bareinboim, 2024) to existing work in group fairness and robustness. Singh et al. (2021) and Schrouff et al. (2022) further study properties of fair classifiers under specific distribution shifts, with Makar & D'Amour (2022) demonstrating that fairness and robustness may be in alignment under some assumptions on the causal structure of the data.

The group parity paradigm considers fairness as something that can be traded off for performance, whereas the latter two paradigms consider fairness as aligned with maximising subgroup-wise performance on a given test set (either IID or unbiased). For this reason, we will refer to the latter paradigms as *performance-sensitive*. In this work, we ask a simple question: are FRL methods (which were developed for the group parity paradigm) valid when applied in performance-sensitive settings?

## 3 CAUSAL STRUCTURES OF DATASET BIAS

We now take a moment to define what we mean when we say that a dataset is biased. We consider classification problems where we have access to a training dataset of inputs $\mathbf{X}$, targets $Y$, and sensitive attributes $A$. The targets are a potentially noisy reflection of some unobserved underlying condition $Z$ (i.e. $Y := Z$ when there is no label noise). Taking a causal interpretation, let $\mathfrak{C}_{\mathrm{tr}}$ be a structural causal model (SCM) representing the generative processes in the training dataset. Similarly, $\mathfrak{C}_{\mathrm{te}}$ is the SCM of the test dataset on which we want to make predictions. We focus on the task of learning a probabilistic model approximating the conditional test distribution $P^{\mathfrak{C}_{\mathrm{te}}}(Y \mid \mathbf{X})$.

Fair representation learning is predicated on the idea that inputs may encode sensitive subgroup information that may be spuriously correlated with the targets. To express this distinction between task-specific and sensitive information, we need a richer description of our input vector. Following Jiang & Veitch (2022), we consider $\mathbf{X}$ to be a random vector which may be partitioned into two random variables: $X_Z$, representing target-related features directly caused by $Z$; and $X_A$, representing features related to the sensitive attribute, directly caused by $A$.

By construction, $X_A$ is predictive of the sensitive attribute $A$, so we say it encodes *sensitive information*. In high-dimensional problems, such as imaging, we may view $\{X_Z, X_A\}$ as high-level latent features that models may implicitly depend on when trained to predict $Y$ from $\mathbf{X}$. For instance, consider a skin lesion classification task where self-reported race is the sensitive attribute. Here, $X_Z$ may be the pixels representing the lesion, whereas $X_A$ may correspond to skin pigmentation. Importantly, the amount of sensitive information encoded in the inputs may vary across application domains due to differences in the $A \rightarrow X_A$ pathway. Jones et al. (2023) refer to the ease with which $A$ may be predicted from $X_A$ as *subgroup separability*, finding that performance degradation of ERM models under dataset bias is strongly affected by subgroup separability.

When the mapping from inputs to targets is consistent across groups, sensitive information is irrelevant for class prediction; we define such distributions as *unbiased*. In our skin lesion example, the dataset would be unbiased if exploiting skin pigmentation information does not help performance on the lesion classification task. We formalise this notion of dataset bias in Definition 3.1.

**Definition 3.1** (Unbiased distribution)**.** The distribution induced by a structural causal model $\mathfrak{C}$ is unbiased if, given $X_Z$, sensitive information $X_A$ provides no information relevant to predicting $Y$[1]:

$$Y \perp\!\!\!\perp^{\mathfrak{C}} X_A \mid X_Z \iff P^{\mathfrak{C}}(Y \mid X_Z) = P^{\mathfrak{C}}(Y \mid X_Z, X_A).$$

Applying the graphical d-separation criterion (Verma & Pearl, 1990), we may derive three fundamental mechanisms of dataset bias that may violate Definition 3.1, causing sensitive information to become spuriously correlated with the target (Jones et al., 2024). We illustrate the unbiased distribution in Figure 1a and highlight each of the three potential shortcuts in Figure 1{b – d}.

---

[1] $\perp\!\!\!\perp$ represents statistical independence, see §A.1 for a table of notation.

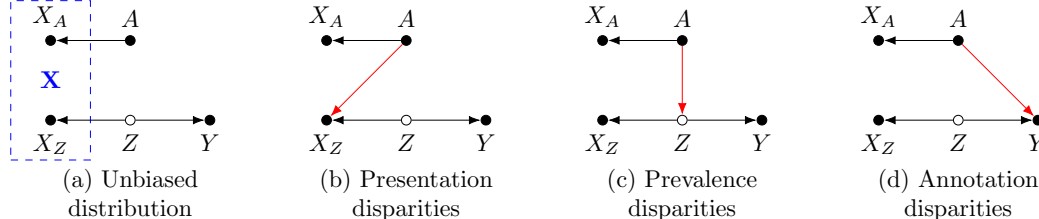

Figure 1: Causal structures of dataset bias in classification tasks. The input $\mathbf{X}$ is decomposed into latent features $X_Z, X_A$ based on their causal relationships with the sensitive attribute $A$ and (unobserved) underlying class $Z$. In the unbiased setting (a), sensitive information is irrelevant to predicting the target $Y$. This condition may be violated by (b) feature entanglement of $A$ and $Z$, (c) differences in base rates across subgroups, or (d) differences in labelling policy across subgroups.

Figure 1b is a disparity in class *presentation* $\exists\,(a, a^*) : P(X_Z \mid Z, a) \neq P(X_Z \mid Z, a^*)$, where the same features encode sensitive and class-specific information. This is in contrast to disparities in class *prevalence* $\exists\,(a, a^*) : P(Z \mid a) \neq P(Z \mid a^*)$ illustrated in Figure 1c, where base rates shift across groups. Finally, Figure 1d represents disparities in *annotation* $\exists\,(a, a^*) : P(Y \mid Z, a) \neq P(Y \mid Z, a^*)$, where different groups are labelled with different policies. These structures represent realistic sources of bias, with Jones et al. (2024) discussing extensively how each may occur naturally in medical imaging scenarios. We provide further background and discussion, with a brief worked example of each mechanism in Appendix §A.2.

We will assume in this paper that the disparities in Figure 1{b–d} are spurious and should be mitigated. However, any of the mechanisms in Figure 1 may constitute fair or unfair situations in the real world (Chiappa, 2019). For example, in medical imaging, disease prevalence and presentation may legitimately vary across populations (Mccradden et al., 2023) due to known physiological mechanisms. In practice, a domain expert would need to determine the fairness of each situation.

## 4 RETHINKING FAIR REPRESENTATIONS

Motivated by our causal formulation of dataset bias, we take a detailed look at the limits of fair representations from the perspective of performance-sensitive fairness paradigms. Let's begin by recalling from Zemel et al. (2013) that the stated aim of fair representation learning is to

> *"lose any information that can identify whether the person belongs to the protected subgroup, while retaining as much other information as possible"*.

We will refer to the first part of this goal as *effectiveness* – is FRL effective at removing sensitive information that would have been encoded by ERM? The second part will be called *harmlessness* – does FRL avoid harming performance by retaining task-relevant information? We begin by proving that fair representations cannot be both effective and harmless if test data is IID to training data.

Notably, our results follow from our causal setup in §3, showing how a causal approach helps to clarify complex issues in bias and fairness. We do not presuppose any architecture or implementation for the classifiers. Nor do we make assumptions about the functional mechanisms in the underlying SCM. We scrutinise the objective of learning fair representations through the lens of implied conditional independence relationships. By taking this approach, we focus on the underlying structure of the distribution being approximated, as opposed to the training dynamics of any specific model. We include a discussion of assumptions and proofs for all Lemmas in §A.3.

### 4.1 FUTILITY IN THE IID PERFORMANCE PARADIGM

**Preliminaries** We consider models of the following form: a feature extractor $f_\theta$ mapping inputs to representations $R$, and a classifier which maps representations to predictions. Both components are typically implemented as (deep) neural networks. Fair representation learning imposes the train-time constraint that fair representations $R^{\mathrm{FRL}}$ must be (marginally) independent of the sensitive attribute,

denoted as $R^{\text{FRL}} \perp\!\!\!\perp^{\mathfrak{C}_{\text{tr}}} A$, leading to a predictor satisfying demographic parity. We contrast this to the unconstrained ERM strategy (i.e. learning $R^{\text{ERM}}$). While $f_\theta$ is always a function of $\mathbf{X}$ (i.e. the feature extractor takes the whole of $\mathbf{X}$ as input), we will slightly abuse the notation $f_\theta(\mathbf{X}^*)$ to indicate that the feature extractor is only non-constant w.r.t. some subset $\mathbf{X}^*$ of $\mathbf{X}$.

**Assumption 4.1.** Unconstrained representations depend on input features $\mathbf{X}^* \subseteq \mathbf{X}$ iff they form a Markov blanket over $Y$ at train-time:

$$R^{\text{ERM}} = f_\theta(\mathbf{X}^*) \iff Y \perp\!\!\!\perp^{\mathfrak{C}_{\text{tr}}} (\mathbf{X} \setminus \mathbf{X}^*) \mid \mathbf{X}^*. \tag{1}$$

The Markov blanket contains all information sufficient to predict $Y$ in an idealised (infinite-sample) setting (Peters et al., 2017, Chapter 6). We may view $\mathbf{X}^*$ as a sufficient statistic for predicting $Y$; hence Assumption 4.1 is closely related to the information bottleneck principle (Tishby et al., 2000), which stipulates that representations should be minimal and sufficient for predicting $Y$. Intuitively speaking, Assumption 4.1 states that a properly trained ERM model encodes relevant information in its representations whilst ignoring irrelevant information.

**Lemma 4.2.** *Fair representations must depend on $X_Z$ only:*

$$R^{\text{FRL}} \perp\!\!\!\perp^{\mathfrak{C}_{\text{tr}}} A \implies R^{\text{FRL}} = f_\theta(X_Z). \tag{2}$$

**Lemma 4.3.** *Unconstrained representations are fair iff the training distribution is unbiased:*

$$R^{\text{ERM}} \perp\!\!\!\perp^{\mathfrak{C}_{\text{tr}}} A \iff Y \perp\!\!\!\perp^{\mathfrak{C}_{\text{tr}}} X_A \mid X_Z. \tag{3}$$

We now take an information-theoretic perspective to define our two desiderata for fair representations: effectiveness (Definition 4.4), and harmlessness (Definition 4.5). While both properties are intuitive and desirable, we show how they each imply constraints on the training and testing distributions in Lemmas 4.6 and 4.7, respectively. By showing that these constraints are incompatible when the distributions coincide, we derive our futility result for IID settings (Proposition 4.8). We denote $I^{\mathfrak{C}}(\cdot; \cdot)$ the mutual information between random variables in the distribution induced by $\mathfrak{C}$.

**Definition 4.4** (Effectiveness). Fair representations are effective if, at train-time, they do not encode sensitive information that unconstrained representations would encode:

$$I^{\mathfrak{C}_{\text{tr}}}(A; R^{\text{ERM}}) > I^{\mathfrak{C}_{\text{tr}}}(A; R^{\text{FRL}}) = 0. \tag{4}$$

**Definition 4.5** (Harmlessness). Fair representations are harmless if, at test-time, they have equal information relevant to predicting the targets as the input (i.e. they do not discard relevant information).

$$I^{\mathfrak{C}_{\text{te}}}(Y; R^{\text{FRL}}) = I^{\mathfrak{C}_{\text{te}}}(Y; X_Z, X_A). \tag{5}$$

**Lemma 4.6.** *Effectiveness ($\mathcal{E}$) implies bias at train-time:*

$$\mathcal{E} \implies Y \not\perp\!\!\!\perp^{\mathfrak{C}_{\text{tr}}} X_A \mid X_Z. \tag{6}$$

Intuitively, Lemma 4.3 implies that an unconstrained model will not encode sensitive information in its representations when trained on a dataset where that information is irrelevant for task prediction (i.e. unbiased according to Definition 3.1). However, effectiveness (Definition 4.4) requires that an unconstrained model does encode sensitive information in its representations – else there would be no point removing it with FRL! Thus, Lemma 4.6 follows, stating that FRL can only be effective if the training data is biased.

**Lemma 4.7.** *Harmlessness ($\mathcal{H}$) implies no bias at test-time:*

$$\mathcal{H} \implies Y \perp\!\!\!\perp^{\mathfrak{C}_{\text{te}}} X_A \mid X_Z. \tag{7}$$

Lemma 4.7 states that enforcing demographically invariant representations must lead to a performance penalty when testing on a biased (according to Definition 3.1) dataset. This result is closely related to Zhao & Gordon (2019), who relate the performance penalty under prevalence disparities to the difference in base rates across groups. Our result in Lemma 4.7 does not attempt to derive any bounds on the performance penalty, but is more general. We show that there is a performance penalty when deploying FRL on *any* dataset violating the unbiasedness condition in Definition 3.1, including (but not limited to) the causal structures in Figure 1{b–d}.

**Proposition 4.8** (Futility). *Fair representations may not be effective ($\mathcal{E}$) and harmless ($\mathcal{H}$) if the train and test datasets are identically distributed:*

$$\mathcal{E} \wedge \mathcal{H} \implies P^{\mathfrak{C}_{tr}} \neq P^{\mathfrak{C}_{te}}. \tag{8}$$

*Proof.* Suppose, for the sake of contradiction, that we have IID training and testing distributions $P^{\mathfrak{C}_{tr}} = P^{\mathfrak{C}_{te}}$ and that effectiveness and harmlessness are satisfied. Substituting Lemmas 4.6 and 4.7, we get that

$$\mathcal{E} \wedge \mathcal{H} \implies (Y \not\perp\!\!\!\perp X_A \mid X_Z) \wedge (Y \perp\!\!\!\perp X_A \mid X_Z),$$

which is a contradiction. □

We emphasise the importance of Proposition 4.8, given that performance-oriented IID benchmarks persist in the literature. *Fair representation learning is futile for performance-sensitive IID tasks.* The strategy carries an implicit assumption that training data contains bias not present at test time. Intuitively, preventing a model from using information can only worsen performance unless the predictive power given by that information is entirely spurious and expected to disappear at test time. Proposition 4.8 hence provides a theoretical explanation on why previous empirical studies benchmarking FRL methods in IID settings did not find any consistent improvements over ERM methods (Pfohl et al., 2021; Zhang et al., 2022; Zietlow et al., 2022; Zong et al., 2023).

## 4.2 POTENTIAL VALIDITY IN THE DISTRIBUTION SHIFT PARADIGM

Proposition 4.8 demonstrates that FRL methods cannot be motivated by performance in IID settings. We now turn our attention to whether FRL may benefit performance under distribution shift. This setting is more interesting, and today, contradictory empirical results exist in the literature. For example, Kim et al. (2019) and Tartaglione et al. (2021) demonstrate successes of FRL on simple colour-MNIST benchmarks. In contrast, Wang et al. (2020) find that FRL methods fail on the more complex CIFAR-S benchmark. Such results seem to indicate that the underlying structure of the dataset and the shift may affect the validity of FRL methods in the distribution shift paradigm.

To minimise test-time risk under distribution shift, we need some notion of what information is stable across domains and what information is unstable or spurious (Peters et al., 2016; Arjovsky et al., 2019). Revisiting Figure 1, notice that while the shortcut paths (red arrows) are unstable across domains, the $Z \to X_Z$ causal pathway is stable across all causal structures, and thus an encoder which depends only on $X_Z$ is necessary to transport from a biased training setting to an unbiased deployment setting (Jiang & Veitch, 2022; Makar & D'Amour, 2022). This is encouraging for FRL, as Lemma 4.2 demonstrates that depending on $X_Z$ only is a necessary condition for fair representations.

Crucially, however, Lemma 4.2 is not a sufficient condition. There is no guarantee that enforcing $R^{FRL} \perp\!\!\!\perp^{\mathfrak{C}_{tr}} A$ is sufficient to learn an encoder which can recover faithful representations of $X_Z$ in all cases. Indeed, there is evidence in the existing literature that the validity of enforcing invariant representations is dependent on the underlying causal structure of the problem, with Veitch et al. (2021) and Makar & D'Amour (2022) each proving results for the robustness of closely related methods under different causal structures of distribution shift.

From a representation learning perspective, proving the validity of FRL would involve proving causal identifiability (Khemakhem et al., 2020) of the $X_Z$ feature, which is challenging in the general case (Hyvärinen et al., 2024). Additionally, even if FRL cannot guarantee identifiability, FRL may still provide a performance benefit over ERM, especially on datasets with a strong bias or high subgroup separability. In such cases, ERM models are more likely to rely on the bias shortcut and may suffer extreme performance degradation (Jones et al., 2023). Given these challenges with theoretical analysis, we focus instead on a simpler and weaker concept of validity: does FRL practically attain better performance than ERM? We propose two hypotheses, which we will explore in §5.

> **Hypothesis 4.9.** FRL *validity under distribution shift depends on the underlying causal structure of the bias present at train-time.*
>
> **Hypothesis 4.10.** FRL *validity under distribution shift depends on the amount of sensitive information initially present in the inputs (subgroup separability).*

## 5 EXPERIMENTS AND RESULTS

We support our theoretical analysis with a large-scale set of experiments on medical image data. We adapt the experimental setup from Jones et al. (2023), consisting of five datasets across the modalities of chest X-ray (CheXpert, MIMIC; Irvin et al., 2019; Johnson et al., 2019), dermatoscopy (HAM10000, Fitzpatrick17k; Tschandl et al., 2018; Groh et al., 2021; Groh et al., 2022), and fundus imaging (PAPILA; Kovalyk et al., 2022). Each dataset is associated with a binary disease classification task and binary sensitive attribute. Where datasets have multiple sensitive attributes available, they are treated separately, giving eleven dataset-attribute combinations. We treat the unaltered datasets as unbiased and generate biased variants of each dataset according to the mechanisms in Figure 1. In each bias mechanism, we inject bias into one subgroup ('Group 1') by either dropping samples, corrupting the image, or corrupting the label, whereas the other subgroup ('Group 0') is left uncorrupted. We provide details on each dataset, including the procedures to generate each biased variant, in §A.4.

Our experiments compare subgroup-wise accuracy of ERM against a popular adversarial FRL method (Kim et al., 2019) and we repeat our analysis with a class-conditional FRL method (Zhao et al., 2020) in §A.6. Both methods are representative of state-of-the-art in FRL[2]. For each dataset-attribute combination, we train each method on each dataset variant over five random seeds for a total of 660 training runs. We evaluate performance by considering the percentage-point difference in mean accuracy between FRL and ERM ($\Delta$ Acc) for each subgroup. For subgroup separability, we use the measurements reported for each dataset by Jones et al. (2023)[3]. Further hyperparameter, training, and model details can be found in §A.5.

### 5.1 VERIFYING FUTILITY IN THE IID PERFORMANCE PARADIGM (PROPOSITION 4.8)

Figure 2 plots the performance gap between FRL and ERM in the IID case. The training and testing datasets are generated by randomly splitting the unbiased variant of each dataset. For all dataset–attribute combinations, $\Delta$ Acc is negative or approximately zero for both subgroups, supporting the finding in Proposition 4.8 that FRL can only maintain or worsen performance in IID settings.

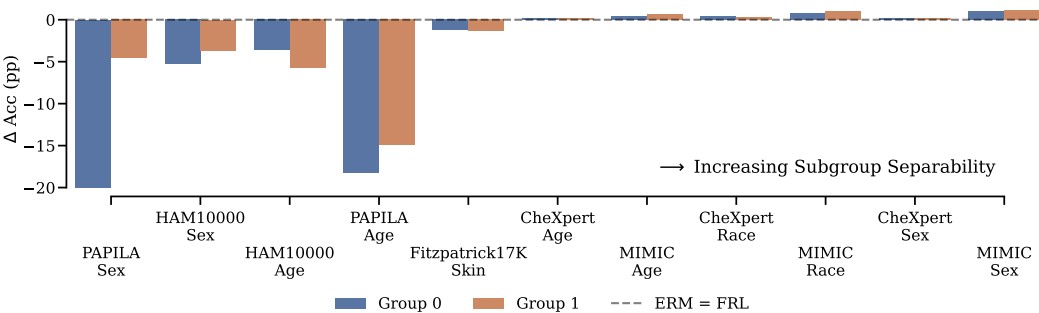

Figure 2: Percentage-point mean accuracy gap for FRL models compared to ERM models on IID disease classification tasks (train/test unbiased). Positive $\Delta$ Acc means FRL outperforms ERM. Datasets are sorted by increasing subgroup separability on the x-axis.

---

[2]Benchmarking by (Zong et al., 2023) found no statistically meaningful differences between FRL methods.

[3]Jones et al. (2023) acquire subgroup separability measurements using test-time AUC of classifiers trained to predict the sensitive attribute. Since we use the same model class – ResNet18 (He et al., 2016) – these measurements are also appropriate for our experiments.

Interestingly, the dataset–attribute combinations which suffered the most under FRL had the lowest subgroup separability, whereas the settings with better FRL performance had higher subgroup separability. This seems to indicate that when inputs encode sensitive information more strongly, FRL is better at removing it without affecting the primary task. Conversely, when sensitive information is more difficult to extract from the inputs, features relevant to the primary task may be more tightly entangled with those relevant to predicting sensitive attributes. In this case, attempting to remove features predictive of the sensitive attribute may degrade primary task performance more.

## 5.2 TESTING POTENTIAL VALIDITY UNDER CAUSAL SHIFTS (HYPOTHESIS 4.9)

We now consider the performance of FRL under distribution shift, testing Hypothesis 4.9 that FRL performance depends on the underlying causal structure of the shift. Figure 3 plots the performance gap between FRL and ERM when trained on each bias mechanism and tested on an unbiased test set. We find that FRL performs best relative to ERM under presentation disparities, where it can boost performance for Group 1 (the disadvantaged group) in settings with high subgroup separability. In the other two bias mechanisms, FRL provides little benefit, providing evidence that the underlying causal structure of the bias matters for the practical validity of FRL.

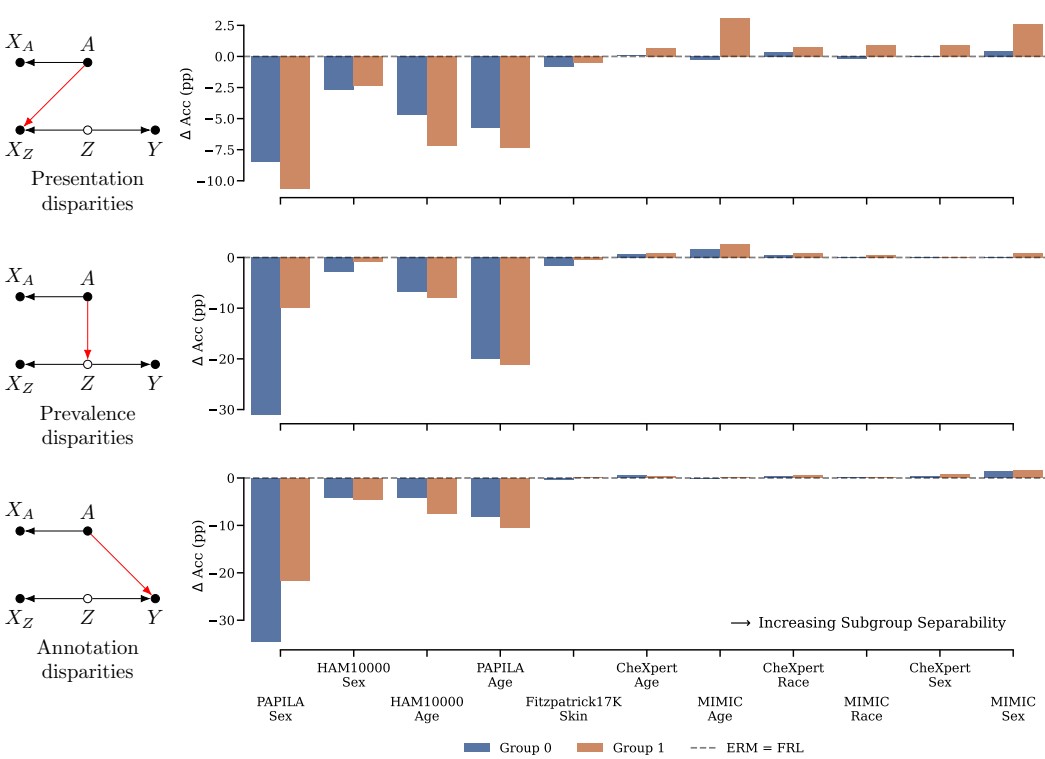

Figure 3: Percentage-point mean accuracy gap for FRL models compared to ERM models when trained on each mechanism of dataset bias (test set is always unbiased). Positive Δ Acc indicates that FRL outperforms ERM on the unbiased test set.

## 5.3 TESTING POTENTIAL VALIDITY AS A FUNCTION OF SUBGROUP SEPARABILITY (HYPOTHESIS 4.10)

Perhaps the most noticeable pattern in Figure 3 is how the performance gap varies strongly with separability, supporting Hypothesis 4.10 that the practical validity of FRL depends on subgroup separability. Across all bias mechanisms, FRL did not offer any improvements for datasets with low subgroup separability, similar to what has been observed in the unbiased settings. We investigate this further in Figure 4, which aggregates the results from Figure 3 over all three bias mechanisms, using

the subgroup separability AUC from Jones et al. (2023) as the x-axis. Our results indicate that there is clear correlation between subgroup separability and empirical validity of FRL.

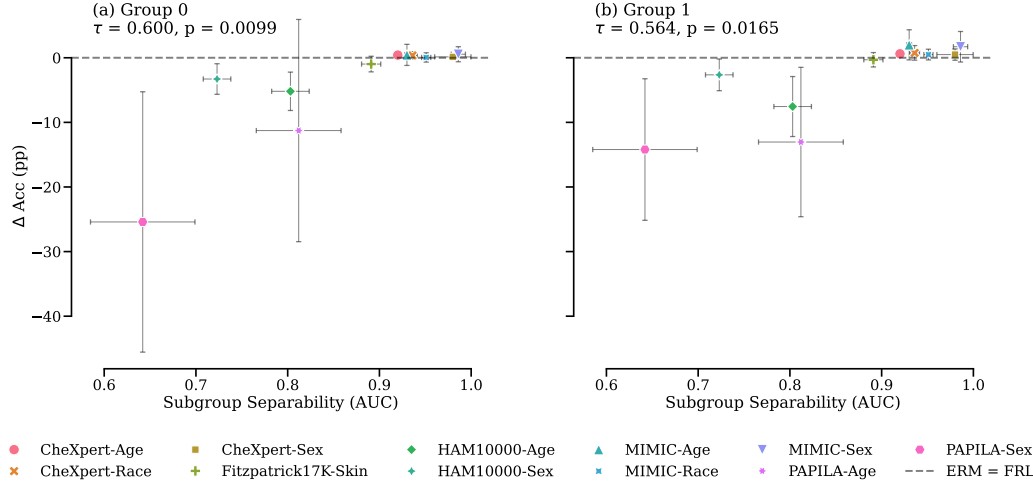

Figure 4: Percentage-point mean accuracy gap for FRL models compared to ERM models, aggregated over all bias mechanisms and plotted against subgroup separability AUC, as reported by Jones et al. (2023). Positive $\Delta$ Acc indicates that FRL outperforms ERM on the unbiased test set. We use Kendall's $\tau$ statistic to test for a monotonic association between $\Delta$ Acc and subgroup separability. $y$-axis error bars represent standard deviations of the aggregated $\Delta$ Acc measurements. $x$-axis error bars represent standard deviations in subgroup separability measurements.

Figure 4 makes the dependence of FRL validity on subgroup separability clear, demonstrating a statistically significant monotonic association between $\Delta$ Acc and subgroup separability. On dataset-attribute combinations with high subgroup separability, FRL improves performance relative to ERM for the disadvantaged group (Group 1) whilst maintaining performance for other the group. In settings with low separability, FRL substantially worsens performance for both groups.

## 6 DISCUSSION

By organising the related literature into three paradigms of fairness analysis in §2, our work helps to untangle confusion across previous work stemming from multiple conflicting evaluation paradigms and implicit assumptions about what is considered fair. Our causal treatment of dataset bias in §3 shines a light on how the structure of the underlying distribution is key to reasoning about fairness, directly motivating our theoretical and empirical results in §4 and §5. We discuss three insights from our work and potential directions for the field.

**FRL is not a useful fairness strategy for performance-sensitive IID tasks** Proposition 4.8 states that if we are to apply fair representation learning on IID benchmarks, we must implicitly drop one of the effectiveness or harmlessness criteria. Which criterion we lose depends on whether our data is biased or unbiased according to Definition 3.1.

On unbiased data, Lemma 4.6 shows that we must drop the effectiveness criterion, so FRL provides no fairness benefit over ERM, which would not encode sensitive information anyway (provided Assumption 4.1 holds, as discussed in §A.3). Furthermore, there is no reason for one to implement FRL (or indeed any bias mitigation method) if they were confident that they had an unbiased dataset.

On biased data, Lemma 4.7 shows how we lose the harmlessness criterion and should expect overall test-time performance to degrade relative to ERM methods. One interpretation of this result is that group fairness metrics such as demographic parity are not aligned with minimax fairness (Martinez et al., 2020) under dataset bias; thus, Lemma 4.7 may be seen as a general impossibility result. It is complementary to Pfohl et al. (2023), who investigate whether Bayes-optimal classifiers satisfy equalised odds under causal structures of dataset bias. Note that Lemma 4.7 does not provide

bounds for the amount of performance degradation; these may be derived for narrower settings with assumptions on the bias mechanisms (Zhao & Gordon, 2019; Zhao et al., 2022).

In this light, recent results from real-world evaluations (e.g. Pfohl et al., 2021; Zhang et al., 2022; Zietlow et al., 2022; Zong et al., 2023), showing that FRL methods worsen performance for all groups, are unsurprising and may be viewed as fairness-performance tradeoffs. Real-world datasets typically have some amount of pre-existing bias, and most evaluations are IID because the train/test sets are generated via random splitting. We should not expect FRL methods to achieve state-of-the-art performance in these cases, and we caution against enforcing invariant representations if evaluation and deployment settings are expected to be IID to training. FRL methods should not be used 'blindly'.

**Statistical and causal considerations affect the validity of FRL under distribution shift** By taking a fine-grained approach, our work proposes – and provides empirical evidence for – two statistical and causal factors that are rarely considered in fairness analysis (Hypotheses 4.9 & 4.10).

Our results in §5 demonstrate how the empirical validity of FRL in the distribution shift paradigm depends on both the causal structure of the bias *and* the amount of sensitive information present to begin with (subgroup separability). We found that FRL methods could only improve performance on an unbiased test set relative to ERM when trained on datasets with presentation disparities and high subgroup separability. When trained on other bias mechanisms or on data with lower subgroup separability, FRL consistently degraded performance relative to ERM. Particularly notable was the magnitude of the performance degradation as subgroup separability decreased.

We argue that further theoretical work to understand the precise relationship between dataset bias, subgroup separability, and generalisation performance of ERM and FRL under distribution shift will be a particularly productive area of study moving forward. We provide an extended discussion on connections to domain generalisation and potential directions for future work in §A.2.

**Real-world evaluation of FRL remains challenging** Finally, we emphasise that real-world evaluation of fairness methods under the distribution shift paradigm remains a challenge. Proper evaluation of FRL under distribution shift requires training on a biased dataset and testing on an unbiased one, but it is tough to find real-world data which satisfy these criteria; we rarely have full knowledge of the biases, and if we had access to an unbiased dataset, we could use it for training without needing FRL.

To overcome this obstacle, some work (e.g. Kim et al., 2019; Tartaglione et al., 2021) leverages synthetic data with known biases. Others (e.g. Wang et al., 2020, and our experiments in §5) take the alternative approach of injecting bias into real-world data. However, both approaches are unlikely to perfectly simulate the true complexity of real-world biases. Until we better understand the causal and statistical nature of real-world bias, proper evaluation of fairness methods will remain difficult. Other disciplines have a long history of using standardised research protocols and reporting guidelines (e.g. for clinical trials). It may be time to consider similar strategies for planning and assessing research advances on the frontiers of machine learning.

ACKNOWLEDGEMENTS

C.J. is supported by Microsoft Research, EPSRC, and The Alan Turing Institute through a Microsoft PhD scholarship and a Turing PhD enrichment award. M.R. is supported by an Imperial College London President's PhD scholarship and a Google PhD fellowship. B.G. received support from the Royal Academy of Engineering as part of his Kheiron/RAEng Research Chair. B.G. and F.R. acknowledge the support of the UKRI AI programme, and the Engineering and Physical Sciences Research Council, for CHAI - EPSRC Causality in Healthcare AI Hub (grant number EP/Y028856/1).

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

# A APPENDIX

## A.1 ACRONYMS AND NOTATION

| | |
|---|---|
| **IID** | independent and identically distributed. |
| **ERM** | empirical risk minimisation. |
| **SCM** | structural causal model. |
| **FRL** | fair representation learning. |

| | |
|---|---|
| $X$ | random variable. |
| $\mathbf{X}$ | random vector. |
| $x$ | scalar realisation of random variable $X$. |
| $\mathbf{x}$ | vector realisation of random vector $\mathbf{X}$. |
| $\mathfrak{C}$ | structural causal model. |
| $P^{\mathfrak{C}}$ | probability distribution induced by $\mathfrak{C}$. |
| $X \perp\!\!\!\perp^{\mathfrak{C}} Y$ | $X, Y$ are statistically independent in the distribution induced by $\mathfrak{C}$. |
| $I^{\mathfrak{C}}(X;Y)$ | mutual information between $X, Y$ in the distribution induced by $\mathfrak{C}$. |

## A.2 EXTENDED DISCUSSION

We provide an extended discussion, adding depth to areas that some readers – particularly those interested in building on this work – may find interesting. Some elements of this section are adapted from conversations during the review process. We thank the anonymous reviewers for spurring us to think about these topics.

### WORKED EXAMPLES OF BIAS MECHANISMS

To further illustrate the bias mechanisms in §3, consider a medical example where we wish to classify some disease $Y$ from chest X-ray images $X$, with biological sex as a sensitive attribute:

- A prevalence disparity may take the form of a shift in the marginal distribution of $Y$ across groups. For example, there may be a greater proportion of positive males in the dataset than positive females due to some combination of physiological differences, demographic differences, historical disparities in healthcare, etc.

- A presentation disparity may occur if there is a shift in the generative process $P(\mathbf{X} \mid Y)$; for example, one group may be systematically diagnosed later in their disease progression, leading to the same condition appearing more severe or with different pathological features.

- An annotation disparity is when there is a shift in the diagnostic mapping $P(Y \mid Z)$. This may occur if different groups are annotated with different policies, e.g. due to historical healthcare disparities or diagnosis practices at different hospitals.

Each of these mechanisms would cause an ERM-trained classifier to rely on sensitive information when trained to predict disease from chest X-rays. Importantly, for all cases, a domain expert would need to examine the causes of each disparity. If the association is deemed spurious or unfair (as we assume throughout this paper), then the disparity should be mitigated. If this association is not spurious, then it may contain potentially useful information that a predictive model should leverage.

### ON THE CHOICE OF CAUSAL STRUCTURES

The bias mechanisms in Figure 1 are derived by applying the d-separation criterion (Verma & Pearl, 1990) to find the simplest fundamental graphs which violate Definition 3.1. They are based on the causal structures proposed by Jones et al. (2024), who also provide practical examples justifying their applicability to real-world settings. Notably, when there is no possibility of label noise (i.e. $Y := Z$), these structures collapse to the familiar anticausal setup (i.e. $\mathbf{X} \leftarrow Z \rightarrow Y$ becomes $\mathbf{X} \leftarrow Y$).

These structures may thus be seen as generalisations of previously studied anticausal bias mechanisms, such as those from Singh et al. (2021) and Makar & D'Amour (2022), and so results that are valid for our setup should be valid for the anticausal case in general. Similar structures have also been applied in the robustness and distribution shift literature (Veitch et al., 2021; Jiang & Veitch, 2022).

We also note that, while we use the structures in Figure 1 in our setup and to motivate the biases in §5, our futility result in Proposition 4.8 is more general. Proposition 4.8 relies only on Definition 3.1 and the causal decomposition of $\mathbf{X}$ into $\{X_A, X_Z\}$. We thus emphasise that the class of problems for which FRL is futile is much larger than the causal structures explicitly enumerated in §3. See Figure 5 for two further relevant examples.

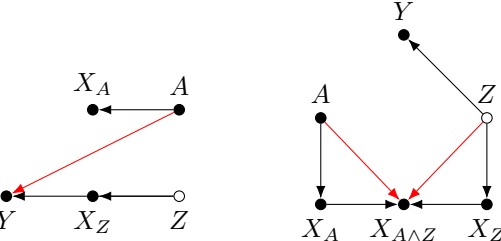

Figure 5: Two further examples of bias mechanisms for which Proposition 4.8 applies to. Left is a causal structure (i.e. $\mathbf{X} \to Y$), where different groups with the same $X_Z$ features are annotated differently. Right includes an interaction feature $X_{A \wedge Z}$, acting as a collider for $A$ and $Z$. Any model that implicitly conditions on the $X_{A \wedge Z}$ feature will see a spurious correlation between $A$ and $Z$.

CONNECTIONS TO COUNTERFACTUAL FAIRNESS

Our work focuses on statistical notions of performance and fairness, as these are most commonly evaluated by the community and are explicitly targeted by FRL methods. This is closely related to causal notions of fairness such as counterfactual fairness (Rosenblatt & Witter, 2022; Anthis & Veitch, 2024), however, there are some subtleties to this connection (Silva, 2024). In many cases – but notably, not all (Silva, 2024) – counterfactual fairness implies demographic parity, and in such situations, Proposition 4.8 also applies to counterfactually fair FRL predictors. Similarly, conditional FRL enforces equal opportunity, which can be implied by extensions of counterfactual fairness such as path-specific counterfactual fairness (Chiappa, 2019), allowing similar results to be derived.

CONNECTIONS TO DOMAIN GENERALISATION

On one level, FRL and domain generalisation/adaptation techniques (especially methods based on adversarial training and disentanglement) share many similarities. Often, the problem setup in both of these fields is very similar, with fairness using a 'sensitive attribute' and domain generalisation using a 'domain' or 'environment' variable. This similarity gives us hope that our results may be insightful in fields beyond fairness, however, we do not consider such claims within the scope of this paper.

A key point to note is that in our formulation, there are two simultaneous shifts: a disparity across groups (e.g. prevalence, presentation, or annotation disparities, as enumerated in Figure 1) and a potential shift across train/test domains (i.e. training on biased data and testing on unbiased data in the distribution shift paradigm). We can thus relate our problem to the traditional distribution shift setup by extending the framework of Federici et al. (2021) for instance.

To illustrate this, consider the joint distribution over training and testing datasets by using the binary indicator variable $T$ to distinguish between them. We can now decompose the shift across groups and domains like so:

$$I(\mathbf{X}, Y, A; T) = I(\mathbf{X}; T) + I(Y; T \mid \mathbf{X}) + I(A; T \mid \mathbf{X}, Y). \tag{9}$$

In this formulation, the LHS term represents the overall distribution shift, and the terms on the RHS represent covariate shift, label shift, and attribute shift, respectively. If the selection yields no information about the joint distribution then the training and test distributions are IID, i.e. $I(\mathbf{X}, Y, A; T) = 0$. Different factorisations of this joint mutual information imply different data-generating processes

and correspond to the various shifts shown in Figure 1. Furthermore, different selection effects can be represented by the functional relation between $(\mathbf{X}, Y, A)$ and the selection variable $T$.

For instance, an attribute-based selection effect could be represented by the causal mechanism $T := f(A, N)$, where $N$ is an exogenous noise variable. There are various other combinations possible, including multivariate selection effects $T := f(\mathbf{X}, Y, A, N)$, or ones consisting only of exogenous noise $T := f(N)$, which would represent the unbiased setting.

Federici et al. (2021) derive some practical upper bounds on the (latent) concept shift quantity which, under some reasonable assumptions, are guaranteed to minimise concept shift. In our view, deriving practical bounds for other types of distribution shifts of the sort studied in the present work and beyond constitutes fertile ground for future research.

### A.3 PROOFS AND DISCUSSION OF ASSUMPTIONS

**Lemma 4.2.** *Fair representations must depend on $X_Z$ only:*

$$R^{\mathrm{FRL}} \perp\!\!\!\perp^{\mathfrak{C}_{\mathrm{tr}}} A \implies R^{\mathrm{FRL}} = f_\theta(X_Z). \tag{2}$$

*Proof.* The result follows from the causal decomposition in §3, where $X_A \not\!\perp\!\!\!\perp A$ by definition. Now, let $\mathbf{X} = \{X_Z, X_A\}$, $\mathbf{X}^* \subseteq \mathbf{X}$, and $R^{\mathrm{FRL}} = f_\theta(\mathbf{X}^*)$:

$$R^{\mathrm{FRL}} \perp\!\!\!\perp^{\mathfrak{C}_{\mathrm{tr}}} A \implies X_A \notin \mathbf{X}^* \implies \mathbf{X}^* \subseteq \mathbf{X} \backslash \{X_A\} \implies R^{\mathrm{FRL}} = f_\theta(X_Z). \qquad \square$$

**Lemma 4.3.** *Unconstrained representations are fair iff the training distribution is unbiased:*

$$R^{\mathrm{ERM}} \perp\!\!\!\perp^{\mathfrak{C}_{\mathrm{tr}}} A \iff Y \perp\!\!\!\perp^{\mathfrak{C}_{\mathrm{tr}}} X_A \mid X_Z. \tag{3}$$

*Proof.* This result is a straightforward consequence of our causal decomposition in §3, combined with Assumption 4.1. Let $\mathbf{X} = \{X_Z, X_A\}$, and $\mathbf{X}^* \subseteq \mathbf{X}$, s.t. $Y \perp\!\!\!\perp^{\mathfrak{C}_{\mathrm{tr}}} \mathbf{X} \backslash \mathbf{X}^* \mid \mathbf{X}^*$. Through simple manipulation, we get that

$$
\begin{aligned}
Y \perp\!\!\!\perp^{\mathfrak{C}_{\mathrm{tr}}} X_A \mid X_Z &\iff \mathbf{X}^* = \{X_Z\}, \quad \mathbf{X} \backslash \mathbf{X}^* = \{X_A\}; \\
&\iff R^{\mathrm{ERM}} = f_\theta(X_Z) \quad \text{(Assumption 4.1)}; \\
&\iff R^{\mathrm{ERM}} \perp\!\!\!\perp^{\mathfrak{C}_{\mathrm{tr}}} A. \qquad \square
\end{aligned}
$$

**Lemma 4.6.** *Effectiveness ($\mathcal{E}$) implies bias at train-time:*

$$\mathcal{E} \implies Y \not\!\perp\!\!\!\perp^{\mathfrak{C}_{\mathrm{tr}}} X_A \mid X_Z. \tag{6}$$

*Proof.* This result follows from Definition 4.4 and Lemma 4.3. Begin by noticing that Equation (4) implies the following independence statement:

$$\mathcal{E} \implies I^{\mathfrak{C}_{\mathrm{tr}}}(A; R^{\mathrm{ERM}}) > 0 \implies R^{\mathrm{ERM}} \not\!\perp\!\!\!\perp^{\mathfrak{C}_{\mathrm{tr}}} A.$$

Since this is the logical negation of the LHS of Equation (3), it follows that the RHS must also be negated when effectiveness is satisfied due to the logical equivalence of the sides ($\iff$). Thus, effectiveness implies bias at train time. $\qquad \square$

**Lemma 4.7.** *Harmlessness ($\mathcal{H}$) implies no bias at test-time:*

$$\mathcal{H} \implies Y \perp\!\!\!\perp^{\mathfrak{C}_{\mathrm{te}}} X_A \mid X_Z. \tag{7}$$

*Proof.* This result follows from Definition 4.5 and Lemma 4.2. Starting from the definition of harmlessness, decompose the RHS expression using the chain rule of mutual information:

$$
\begin{aligned}
\mathcal{H} &\iff I^{\mathfrak{C}_{\mathrm{te}}}(Y; R^{\mathrm{FRL}}) = I^{\mathfrak{C}_{\mathrm{te}}}(Y; X_Z, X_A), \\
&\iff I^{\mathfrak{C}_{\mathrm{te}}}(Y; R^{\mathrm{FRL}}) = I^{\mathfrak{C}_{\mathrm{te}}}(Y; X_Z) + I^{\mathfrak{C}_{\mathrm{te}}}(Y; X_A \mid X_Z).
\end{aligned}
$$

From Lemma 4.2, recall that $R^{\mathrm{FRL}} = f_\theta(X_Z)$. Now we may apply the data processing inequality $I^{\mathfrak{C}_{\mathrm{te}}}(Y; f_\theta(X_Z)) \le I^{\mathfrak{C}_{\mathrm{te}}}(Y; X_Z)$ and nonnegativity of mutual information to see that an unbiased test set is necessary (but not sufficient) for harmlessness:

$$\mathcal{H} \implies I^{\mathfrak{C}_{\mathrm{te}}}(Y; X_A \mid X_Z) = 0 \implies Y \perp\!\!\!\perp^{\mathfrak{C}_{\mathrm{te}}} X_A \mid X_Z. \qquad \square$$

**What if Assumption 4.1 is violated?** Assumption 4.1 is needed to define what information a properly trained ERM model relies on and is used in the proofs of Lemmas 4.3 and 4.6. If we reject Assumption 4.1, we get the (rather unintuitive) result that an ERM model may rely on sensitive information even when trained on an unbiased dataset where such information provides no predictive power. In practice, this may occur if the model is underfit or has insufficient training data. In this case, the FRL strategy may have some use for unbiased IID settings. By constraining the solution space, it may be possible for FRL to improve sample efficiency during training, analogous to a regulariser or inductive prior. It is unclear, however, whether this scenario is particularly relevant with today's practice of high-capacity models trained to convergence on large datasets (Zhang et al., 2016).

**Why the strict inequality in Definition 4.4?** Applying Assumption 4.1, we can derive that $I^{\mathfrak{C}_{tr}}(A; R^{\mathrm{ERM}}) = 0 \iff Y \perp\!\!\!\perp^{\mathfrak{C}_{tr}} X_A \mid X_Z$, that is, unconstrained representations encode no sensitive information if and only if the training data is unbiased. In this case, we define FRL as (trivially) ineffective since it cannot provide any fairness benefit over ERM, which would not encode sensitive information anyway. This case is unlikely to be particularly common since it is unclear why any researcher would apply FRL methods to a dataset that they are confident is unbiased.

## A.4 DATASET DETAILS

The datasets were all preprocessed and split using the same procedure as (Jones et al., 2023), who also report summary statistics. For each dataset, the disease prediction task was constructed by binning all available disease labels (e.g. pneumonia, glaucoma) into the positive class. Other labels (e.g. no-finding) were binned into the negative class. Binary subgroup labels for 'Group 0' and 'Group 1' were constructed according to the following criteria:

- When the sensitive attribute is sex: 'Male' = 'Group 0', 'Female' = 'Group 1'.
- For race: 'White' = 'Group 0', 'Non-White' (all other labels) = 'Group 1'.
- For age: $< 60$ = 'Group 0', $\geq 60$ = 'Group 1'.
- For skin type (Fitzpatrick scale): I–III = 'Group 0', IV–VI = 'Group 1'.

To generate the biased variants of each dataset, we implemented the following procedure:

- Presentation disparities: 50% of positive individuals in 'Group 1' have the image corrupted by reducing sharpness[4].
- Prevalence disparities: 50% of positive individuals in 'Group 1' are dropped from the dataset.
- Prevalence disparities: 50% of positive individuals in 'Group 1' are mislabeled as negative.

## A.5 TRAINING DETAILS

Training consisted of two phases: an initial hyperparameter tuning phase, followed by a final sweep with fixed hyperparameters (the latter phase generated the results reported in §5). In the tuning sweep, the methods were trained and evaluated across all datasets. The final hyperparameters were selected by considering combinations for which training successfully converged across all datasets and achieved the best performance. When selecting adversarial coefficients for the two FRL methods, we ensured that the accuracy of the adversarial prediction head did not exceed the approximate prevalence of the subgroups. This was to prevent the selection of hyperparameter values that would result in the adversary being ignored. The final hyperparameters used are reported in Table 1.

## A.6 CONDITIONAL FRL RESULTS

We include the results of our extended experiments using a conditional FRL implementation (Zhao et al., 2020). Notice that the results demonstrate the same trends as the main results in §5.

---

[4] `torchvision==0.18.1` `adjust_sharpness` implementation, with `sharpness_factor` = 0.5.

Table 1: Hyperparameters used across all runs in §5.

| Config | Value |
|---|---|
| Architecture | ResNet18 (He et al., 2016) |
| Optimiser | AdamW (Loshchilov & Hutter, 2018) {lr: $1e-4$, $\beta_1$: 0.9, $\beta_2$: 0.999} |
| Adversarial coefficients | {Marginal FRL: 1.0, Conditional FRL: 0.05} |
| LR Schedule | Constant |
| Max Epochs | 50 |
| Early Stopping | {Monitor: worst group AUC, Patience: 5 epochs} |
| Augmentation | RandomResizedCrop, RandomRotation($15^o$) |
| Batch Size | 256 (32 for PAPILA) |

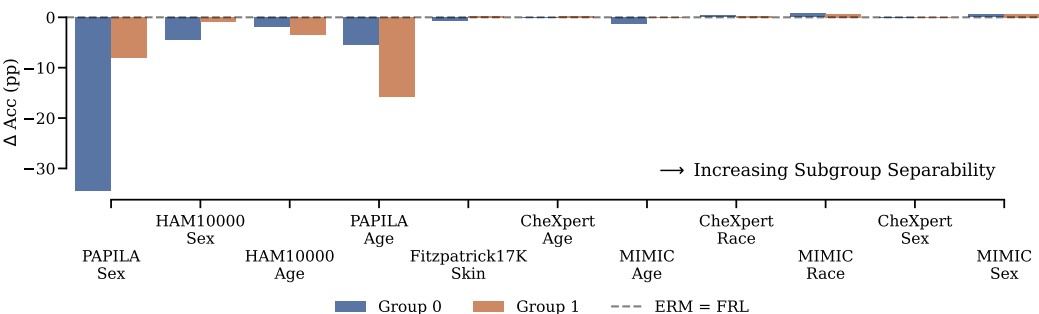

Figure 6: Percentage-point mean accuracy gap for conditional FRL models compared to ERM models on IID disease classification tasks (train/test unbiased). Positive $\Delta$ Acc means FRL outperforms ERM.

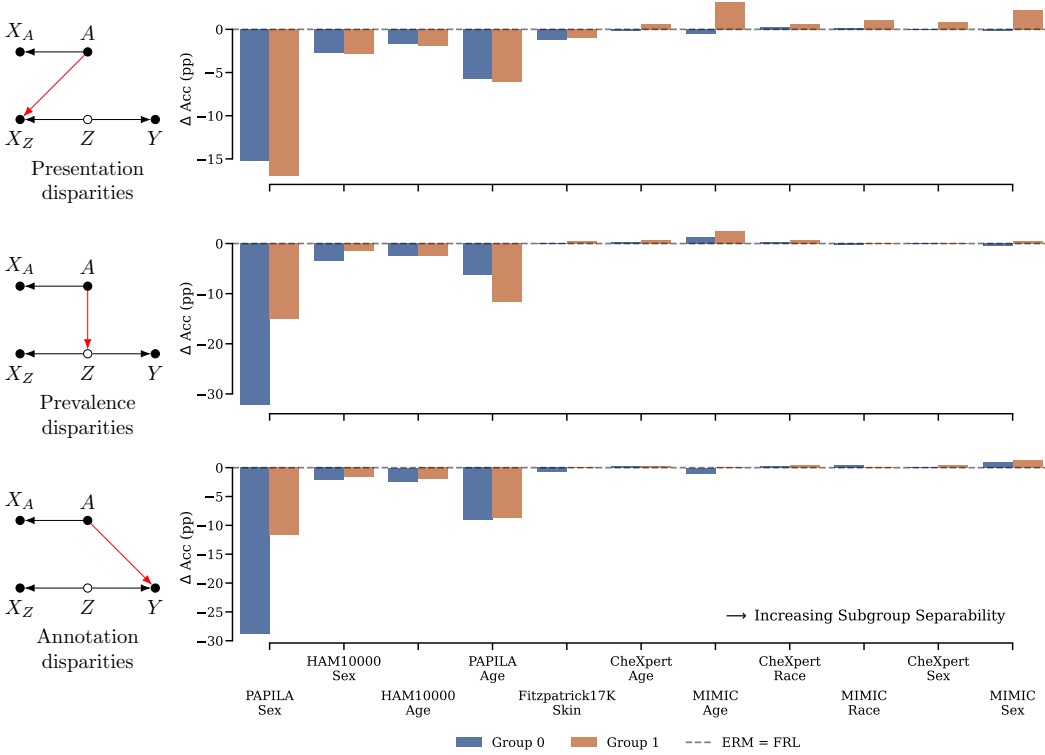

Figure 7: Percentage-point mean accuracy gap for conditional FRL models compared to ERM models when trained on each mechanism of dataset bias (test set is always unbiased). Positive $\Delta$ Acc indicates that FRL outperforms ERM on the unbiased test set.

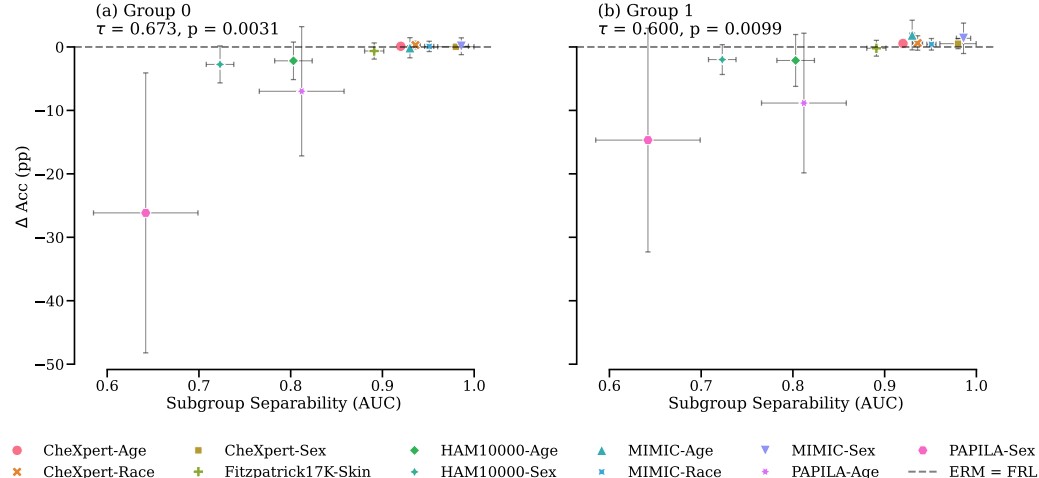

Figure 8: Percentage-point mean accuracy gap for conditional FRL models compared to ERM models, aggregated over all bias mechanisms and plotted against subgroup separability AUC, as reported by Jones et al. (2023). Positive $\Delta$ Acc indicates that FRL outperforms ERM on the unbiased test set. We use Kendall's $\tau$ statistic to test for a monotonic association between $\Delta$ Acc and subgroup separability.

