# OpenReview forum: "Rethinking Fair Representation Learning for Performance-Sensitive Tasks"
_ICLR.cc/2025/Conference — ICLR 2025 Poster_

### Official Review · Reviewer_E1KU · 2024-10-31

**Soundness:** 3
**Presentation:** 3
**Contribution:** 3
**Rating:** 6
**Confidence:** 2

**Summary:**

This paper summarizes the field of fair representation learning into three main categories and then illustrates from a causal perspective that the underlying problem that people have been trying to solve in fair representation learning is in fact somewhat ill-defined. The paper argues that if the test and training distributions are the same, fair representation learning is "futile". In addition, the paper also argues that if there is a distribution shift at test time then there is hope as well as when the features are somewhat separable.
The paper establishes theoretical foundations and well as experimentally illustrates the theoretical findings.

**Strengths:**

- The paper puts into perspective the whole fair representation learning field, using causal language.
- The paper critically establishes a key problem in the field and how researchers benchmark their own methods.
- The paper also shows experimentally that their theoretical findings have practical consequences which is very much appreciated. In particular, they show that if there is indeed a test=train time distribution, fair representation learning does not help much. and vice versa.

**Weaknesses:**

- The paper requires somewhat of a graphical model/ causal background which might be less beginner-friendly
- The paper is very dense and requires multiple passes to fully understand the paper. I would recommend adding further intuitions to the paper.
- Lastly, i might have missed this. Do the authors have any comments on the relations between FRL to standard fairness learning schemes that are not representation-based? I would be curious to hear of these problems persist necessarily.

**Questions:**

see above for the questions

---

> ### Author Response · Authors · 2024-11-20
> **Response to Reviewer Comments**
>
> We thank the reviewer for their feedback and appreciate the positive comments. Please find our responses to the questions below.
>
> > **"The paper requires somewhat of a graphical model/ causal background which might be less beginner-friendly. The paper is very dense and requires multiple passes to fully understand the paper. I would recommend adding further intuitions to the paper."**
>
> We appreciate that this is a complex topic and our analysis relies on techniques that may not be standard for all researchers. To make our work more self-contained and easy to follow, we will add a brief primer on the background topics in the Appendix, including pointers to further introductory material on graphical modelling and causal inference. We will also go through the paper and clarify the trickiest parts with further explanations and intuitions; if there are any parts in particular you would suggest we focus on, please let us know.
>
> > **"Do the authors have any comments on the relations between FRL to standard fairness learning schemes that are not representation-based? I would be curious to hear of these problems persist necessarily.
> "**
>
> We agree that these areas are closely connected. Standard FRL methods are designed to enforce demographic parity on downstream tasks; similarly, conditional FRL methods enforce equalised odds. We note in the discussion that our futility result (Proposition 4.8) may be interpreted as an impossibility result between group fairness and minimax fairness, and we point to [1], who explore a similar concept in the literature.
>
> [1] Pfohl SR, Harris N, Nagpal C, Madras D, Mhasawade V, Salaudeen OE, Heller KA, Koyejo S, D'Amour AN. Understanding subgroup performance differences of fair predictors using causal models. NeurIPS 2023 Workshop on Distribution Shifts: New Frontiers with Foundation Models 2023.

---

> > ### Comment · Reviewer_E1KU · 2024-11-21
> > **Thanks for the reply**
> >
> > I think section 4 and especially 4.1 would benefit from better writing and intuition results of the paper.
> > However, I also concede that given my expertise which is reflected in my confidence score these might be standard in the literature and hence I retain my positive feedback on this paper.

---

### Official Review · Reviewer_nneh · 2024-11-02

**Soundness:** 3
**Presentation:** 3
**Contribution:** 3
**Rating:** 6
**Confidence:** 3

**Summary:**

The paper defines causal structures representing realistic scenarios of dataset bias and discuss how the bias mechanisms may affect the performance and fairness of predictive models. The authors then prove fundamental limitations on fair representation learning when evaluation data is drawn from the same distribution as training data.

**Strengths:**

1. Originality and novelty: the authors propose a novel causal framework to categorize the bias in the data.

2. Supportive theoretical results: the authors prove the limitations of the FRL in IID settings.

3. Clear structure: the writing and organization is clear, making the paper easy to follow.

**Weaknesses:**

1. The causality-based notion in fair ML literature has been well discussed, and there have been numerous causal structures proposed. Can the authors provide a comparative analysis with one or two to further prove the novelty?

2. Limited experiments: it seems the experiments only focus on medical imaging scenarios. The results would be more convincing if datasets from other performance-sensitive areas are considered.

**Questions:**

1. The same as weakness 1, is there more comparison between the causal structures proposed by the authors and other literature?

2. Are there more experiments from another dataset in performance-sensitive areas other than just medical imaging?

3. Typo: in the contribution part (page 2), shouldn't the orderings begin with 1?

---

> ### Author Response · Authors · 2024-11-20
> **Response to Reviewer Comments**
>
> We thank the reviewer for their review and appreciate the comments on the originality and clarity of our work. We respond to the reviewer’s questions and comments below.
>
> > "**The causality-based notion in fair ML literature has been well discussed, and there have been numerous causal structures proposed. Can the authors provide a comparative analysis with one or two to further prove the novelty?**"
>
> The causal structures of dataset bias we consider are derived by applying the d-separation criterion to find the most fundamental graphs that violate Definition 3.1. They are based on the structures proposed in [1], which also provide practical examples justifying their applicability to real-world settings. These structures may be seen as generalisations of previously studied anticausal bias mechanisms, such as those in [2,3]. Similar structures have also been applied in the robustness and distribution shift literature [4,5].
>
> While the structures themselves were used previously, the techniques we use to analyse them and derive new results are novel. Note also that while we use these structures in our setup and experiments, Proposition 4.8 is not limited to only these structures, as it applies to any structure violating Definition 3.1.
>
> We should also clarify that while our structures of dataset bias are causal, our work focuses primarily on statistical notions of (group) fairness, which are weaker than causal notions of fairness [6,7]. While our work has strong connections to this literature, there are some subtleties to this point, which we also discuss in our response to Reviewer svct (question 1). If you feel that this will improve the paper, we are happy to expand on this discussion and add it to the paper.
>
> [1] Jones C, Castro DC, De Sousa Ribeiro F, Oktay O, McCradden M, Glocker B. A causal perspective on dataset bias in machine learning for medical imaging. Nature Machine Intelligence. 2024 Feb;6(2):138-46.
>
> [2] Makar M, D'Amour A. Fairness and robustness in anti-causal prediction. Transactions on machine learning research. 2023 Feb.
>
> [3] Singh H, Singh R, Mhasawade V, Chunara R. Fairness violations and mitigation under covariate shift. Proceedings of the 2021 ACM Conference on Fairness, Accountability, and Transparency 2021 Mar 3 (pp. 3-13).
>
> [4] Jiang Y, Veitch V. Invariant and transportable representations for anti-causal domain shifts. Advances in Neural Information Processing Systems. 2022 Dec 6;35:20782-94.
>
> [5] Veitch V, D'Amour A, Yadlowsky S, Eisenstein J. Counterfactual invariance to spurious correlations in text classification. Advances in neural information processing systems. 2021 Dec 6;34:16196-208.
>
> [6] Kusner MJ, Loftus J, Russell C, Silva R. Counterfactual fairness. Advances in neural information processing systems. 2017;30.
>
> [7] Chiappa S. Path-specific counterfactual fairness. Proceedings of the AAAI conference on artificial intelligence 2019 Jul 17 (Vol. 33, No. 01, pp. 7801-7808).
>
> > "**Limited experiments: it seems the experiments only focus on medical imaging scenarios. The results would be more convincing if datasets from other performance-sensitive areas are considered.**"
>
> We chose to perform our experiments with medical imaging datasets because the field offers a few interesting and important properties for our analysis. First, the domain is certainly performance-sensitive, so it fits well with our scoping of focusing on performance-sensitive tasks. Second, we often see FRL methods applied to medical imaging benchmarks (e.g. [1,2,3]), demonstrating that this is an active and relevant field of research. Third, the medical imaging datasets we use span multiple diverse modalities, with multiple sensitive attributes per dataset and a wide range of subgroup separability, which was especially useful for our analysis in Sections 5.2 and 5.3.
>
> Our experiments cover five datasets over three separate medical modalities, giving eleven dataset-attribute combinations. We agree that more experiments are always better, and we certainly welcome follow-up work in other domains. However, we feel that this is an appropriate scope for this paper, especially considering that our experiments are only one of the contributions of this work. The diversity of the chosen datasets additionally gives confidence that the conclusions are transferable to other domains and applications.
>
> [1] Zong Y, Yang Y, Hospedales T. MEDFAIR: Benchmarking fairness for medical imaging. arXiv preprint arXiv:2210.01725. 2022 Oct 4.
>
> [2] Pfohl SR, Foryciarz A, Shah NH. An empirical characterization of fair machine learning for clinical risk prediction. Journal of biomedical informatics. 2021 Jan 1;113:103621.
>
> [3] Zhang H, Dullerud N, Roth K, Oakden-Rayner L, Pfohl S, Ghassemi M. Improving the fairness of chest x-ray classifiers. InConference on health, inference, and learning 2022 Apr 6 (pp. 204-233). PMLR.

---

> > ### Comment · Reviewer_nneh · 2024-11-20
> > **Re: Response to Reviewer Comments**
> >
> > Thank you for your replies to my questions and I keep feeling positive about this paper. I would suggest including the discussion you provided on comparisons with other causal literature into your revised paper.
> >
> > Regarding the experiments, I agree with you that medical imaging tasks are of great interest, however, I do believe it would be more convincing to include one or two tasks from other fields.

---

### Official Review · Reviewer_ydWf · 2024-11-04

**Soundness:** 4
**Presentation:** 4
**Contribution:** 4
**Rating:** 8
**Confidence:** 5

**Summary:**

This paper makes the argument that fair representation learning (FRL) is not a useful approach in the class of cases where model accuracy (or analogous metrics) is key. They provide an overview of several paradigms of fair ML, and highlighting which are “performance-sensitive”. They then apply a causal framework to discuss the situations in which FRL might be helpful, using this to claim that FRL is not helpful to achieve performance-sensitive objectives when test data is IID. They show experiments to support this, in particular highlighting that even when test data is IID, the causal structure of the problem matters.

**Strengths:**

- Note: I think I may have reviewed a previous version of this paper at another conference (also double blind, no anonymity issues). I have not gone back to look at my past comments so this review is with fresh eyes, and I believe that a number of improvements have been made to this draft. However, apologies if there are overlapping comments anywhere
- Really enjoyed the framing in Sec 2 around group parity, IID performance, and generalization to unbiased OOD, I think this sets up the paper quite well and gives a nice overview of the relevant parts of the fairness literature
- does a nice job of synthetizing causal reasoning with the FRL literature. In particular, the experiments around FRL effectiveness and causal structure are quite interesting
- in general I think the main point made here is correct and correctly-scoped, and useful for a specific segment of the community interested in these things

**Weaknesses:**

- I’m a little confused by the X_Z and X_A framework, what about features that are impacted by both Z and A? It’s not clear if this framework allows for that overlap
- It would be good at some point to discuss whether any of this applies in an anti-causal setup - the “futility” framing is quite strong and yet I feel as though the large anti-causal class of problems are not really touched on here
- Quibble with Def 4.5 - technically, I would phrase this as “have an equal amount of information” rather than “do not discard relevant information”. The question of whether information is discarded I think gets more into implementation and I’m not sure how “discarding information” is defined technically.
- Quibble with Def 4.4/Lemma 4.6 - I find something a little odd here which I think is around the corner case where ERM representations have no sensitive information. I think the important part of Def 4.4 (the “effectiveness”) of FRL is the equality to 0 - this is whether they are truly “fair” in this definition. However, Lemma 4.6 relies heavily on the first part of the def’n (having strictly less information) - if the ERM reprs have 0 sensitive information then trivially all FRL will violate effectiveness, but this seems wrong as that is not a property of the FRL method, but rather a property of the ERM representations which are given. Anyways, this is not a massive issue with the general point but I do think there needs to be some more careful wording with how the intuition is formalized here (e.g. maybe the I(ERM, A)=0 case needs to be separated out as “trivial”)
- The results in Fig 2 on subgroup separability go in the opposite direction of what I would expect - the authors note this and give some thoughts on why this might be. However, it suggests something troubling - that at separability=0.5 (i.e. no information at all about A, trivial case), the fair representation methods will yield low-accuracy representations, when in fact this should be the easiest case (just return the representations as given). This suggests to me that the implemented methods may not be working as intended. It would be good to have a supplementary experiment showing that the methods, are in fact, working as intended - otherwise the empirical results are a little harder to parse due to the chance of experimental failure.

**Questions:**

- The characterization in Fig 1 is interesting but a couple contextual questions - it’s not clear to me as a reader if this is a) a complete characterization of the types of causal structures that can produce this phenomenon, or b) if this is a novel contribution on the part of this paper, or something pre-existing in the literature

---

> ### Author Response · Authors · 2024-11-20
> **Response to Reviewer Comments**
>
> We thank the reviewer for their positive review and sincerely appreciate the constructive comments and depth of engagement with our work. We respond directly to the questions and comments below, clarifying the key points and suggesting improvements we can make to the paper.
>
> >"**I’m a little confused by the $X_Z$ and $X_A$ framework, what about features that are impacted by both $Z$ and $A$? It’s not clear if this framework allows for that overlap.**"
>
> This is an interesting point. In an initial version of this work, we included an interaction feature to explicitly model such situations. However, we realised that this was unnecessary for the proofs in Section 4. This is because the proofs apply to all structures which violate Definition 3.1, of which the three bias mechanisms in Section 3 are simply the most fundamental examples. Furthermore, since the subgraph $A \to X_Z \leftarrow Z$ appears in our formulation of presentation disparities, the interaction case is arguably subsumed by presentation disparities anyway. We will make sure this is clear in the updated version.
>
> Another benefit of this simpler formulation is that it now aligns with the structures proposed by [1], which makes it both easier to justify and more consistent with prior related work.
>
> [1] Jones C, Castro DC, De Sousa Ribeiro F, Oktay O, McCradden M, Glocker B. A causal perspective on dataset bias in machine learning for medical imaging. Nature Machine Intelligence. 2024 Feb;6(2):138-46.
>
> > "**It would be good at some point to discuss whether any of this applies in an anti-causal setup - the “futility” framing is quite strong and yet I feel as though the large anti-causal class of problems are not really touched on here.**"
>
> When there is no possibility of label noise ($Y := Z$), notice that our formulation collapses to the familiar anticausal setup ($X \leftarrow Z \to Y$ becomes $X \leftarrow Y$). In this sense, our formulation is a generalisation of anticausal problems, so our results apply to them too. We will add a note to clarify this.
>
> Indeed, while we use our formulation in the problem setup and the experiments, our futility result is slightly more general (as we noted in our response to your other question). One consequence of this is that we can use our theory to derive even more bias mechanisms for which FRL is futile (e.g.  $Z \to X \to Y$, with $A$ confounding $Y$ and $X$).
>
> >"**Quibble with Def 4.5 - technically, I would phrase this as “have an equal amount of information” rather than “do not discard relevant information”. The question of whether information is discarded I think gets more into implementation and I’m not sure how “discarding information” is defined technically.**"
>
> Thanks for pointing this out, we’re happy to reword this as suggested.
>
> >"**Quibble with Def 4.4/Lemma 4.6 - I find something a little odd here which I think is around the corner case where ERM representations have no sensitive information. I think the important part of Def 4.4 (the “effectiveness”) of FRL is the equality to 0 - this is whether they are truly “fair” in this definition. However, Lemma 4.6 relies heavily on the first part of the def’n (having strictly less information) - if the ERM reprs have 0 sensitive information then trivially all FRL will violate effectiveness, but this seems wrong as that is not a property of the FRL method, but rather a property of the ERM representations which are given. Anyways, this is not a massive issue with the general point but I do think there needs to be some more careful wording with how the intuition is formalized here (e.g. maybe the I(ERM, A)=0 case needs to be separated out as “trivial”)**"
>
> This is an insightful point, and we think we are on the same page about this corner case as you. We thank you for the suggestion to separate this case as trivial; we can add a discussion on this to the Appendix and adapt the main text to signpost it. We think this will help readers who may have similar questions on this part of Definition 4.4.

---

> > ### Author Response · Authors · 2024-11-20
> > **Response to Reviewer Comments (continued)**
> >
> > > "**The results in Fig 2 on subgroup separability go in the opposite direction of what I would expect - the authors note this and give some thoughts on why this might be. However, it suggests something troubling - that at separability=0.5 (i.e. no information at all about A, trivial case), the fair representation methods will yield low-accuracy representations, when in fact this should be the easiest case (just return the representations as given). This suggests to me that the implemented methods may not be working as intended. It would be good to have a supplementary experiment showing that the methods, are in fact, working as intended - otherwise the empirical results are a little harder to parse due to the chance of experimental failure.**"
> >
> > This is a fair question, and we thank you for raising this potential concern. We propose to include a supplementary experiment where we take one or more of the existing datasets and perform some preprocessing which removes as much sensitive information as possible (we can confirm this by training models to predict A from the preprocessed data and ensuring ~0.5 AUC). We can then repeat our experiments on the preprocessed dataset to ensure that the models are working as intended. If these experiments finish on time, we will report back the results here as soon as possible.
> >
> > > "**The characterization in Fig 1 is interesting but a couple contextual questions - it’s not clear to me as a reader if this is a) a complete characterization of the types of causal structures that can produce this phenomenon, or b) if this is a novel contribution on the part of this paper, or something pre-existing in the literature.**"
> >
> > The causal structures of dataset bias we consider are derived by applying the d-separation criterion to find the most fundamental graphs that violate Definition 3.1. They are based on the structures proposed in [1], which also provide practical examples justifying their applicability to real-world settings. As we note in our other response, these structures are useful for formulating the problem and for the experiments, however, our futility result is stronger and applies to any bias mechanism that violated Definition 3.1. We will clarify this in the updated version.
> >
> > [1] Jones C, Castro DC, De Sousa Ribeiro F, Oktay O, McCradden M, Glocker B. A causal perspective on dataset bias in machine learning for medical imaging. Nature Machine Intelligence. 2024 Feb;6(2):138-46.

---

> > > ### Comment · Reviewer_ydWf · 2024-11-26
> > > **Thanks**
> > >
> > > Thanks for the response - I think this paper is good and recommend acceptance, and the proposed additions here sound useful as well.

---

### Official Review · Reviewer_svct · 2024-11-05

**Soundness:** 4
**Presentation:** 4
**Contribution:** 3
**Rating:** 8
**Confidence:** 4

**Summary:**

This paper tackles an important challenge in explainable and reliable AI by examining fair representation learning methods focused on achieving group fairness. The authors critically explore the limitations of existing approaches. They present three main fairness paradigm: enforcing group parity, maximizing subgroup IID performance, and generalizing to unbiased distributions. Their key theoretical insight is that achieving group fairness cannot be both effective and harmless (relative to ERM) within the IID setting. Furthermore, the paper suggests that the success of fairness learning methods depends heavily on shifts in test data distribution, hypothesizing that performance is also influenced by the underlying causal structure and the amount of sensitive information available in the training dataset.

**Strengths:**

This paper is self-contained and easy to follow, and the notation is clear. It is a joyful experience to read a manuscript like this. The theory is simple yet insightful. The experiment supports the theoretical as well.

**Weaknesses:**

1. One major concern is the lack of results in the distributed shift setting. While the negative result in the IID is sound, it would be good to have some results for out-of-distribution, or connects the out-of-distribution fairness to some existed theoretical analysis results in the area of  distribuiton shift.

2. Another concern is that the trade-off between fairness and accuracy has been explored by previous methods (Zhao et al., 2022). Why and how this paper is different than previous methods on analyzing the trade-off between fairness and accuracy,

I would like to raise my score if these concerns are addressed.

[1] Zhao, H., & Gordon, G. J. (2022). Inherent tradeoffs in learning fair representations. Journal of Machine Learning Research, 23(57), 1-26.

**Questions:**

1. This paper mainly talks about group fairness, what about counterfactual fairness? Is the counterfactual fairness also impossible in the iid setting?

2. This work shows that distribution shift is closely related to group fairness, and there are some results in the area of domain generalization / adaptation. Can you elaborate some results in that area? Is there any method closely related to the group fairness?

3. How does different types of distribution shift (class imbalance, covariate shift, concept shift ...) in the dataset related to the three types of disparities?

---

> ### Author Response · Authors · 2024-11-20
> **Response to Reviewer Comments**
>
> We thank the reviewer for their thoughtful feedback and are heartened by the comment that reading our work was ‘a joyful experience’.
>
> Below, we address the concerns and comments point-by-point, with suggested changes to the paper to improve it. We are broadly in agreement on many points and are confident that we can address or at least clarify the main concerns; we welcome any further discussion that we hope will give the reviewer confidence to raise their score.
>
> >"**One major concern is the lack of results in the distributed shift setting. While the negative result in the IID is sound, it would be good to have some results for out-of-distribution, or connects the out-of-distribution fairness to some existed theoretical analysis results in the area of distribution shift.**"
>
> We agree that theoretical results for this setting would be valuable. While there exists some theoretical work deriving results for closely related methods and settings (e.g. [1,2]), it requires stronger assumptions on the data and/or model than what we wanted to make in our paper. Deriving more general results remains a significant challenge for future work.
>
> This motivates the alternative approach we took of proposing two hypotheses on the empirical validity of FRL under distribution shift and using experiments to test them. Our hypotheses and experimental results are novel and potentially surprising as they are somewhat counterintuitive, which we hope will motivate future research in this area. We highlight this point because we believe the scoping and timely contributions of our work are sufficient to be of interest to the community. While adding more results on this aspect is out of scope, we will highlight this as an important next step in the Discussion section.
>
> [1] Jiang Y, Veitch V. Invariant and transportable representations for anti-causal domain shifts. Advances in Neural Information Processing Systems. 2022 Dec 6;35:20782-94.
>
> [2] Makar M, D'Amour A. Fairness and robustness in anti-causal prediction. Transactions on machine learning research. 2023 Feb.
>
> > "**Another concern is that the trade-off between fairness and accuracy has been explored by previous methods (Zhao et al., 2022). Why and how this paper is different than previous methods on analyzing the trade-off between fairness and accuracy.**"
>
> We make distinct and novel contributions that advance the current literature on this topic:
>
> First, by teasing apart three paradigms of fairness analysis in Section 2, we help to expose different assumptions that different studies make about what is considered fair. We feel that this framing will be particularly useful to the community, as it helps to explain why various empirical studies reach seemingly contradictory results. By understanding these distinctions between different paradigms, researchers will be better able to understand and evaluate methods and metrics moving forward.
>
> Second, our causal setup in Section 3 explicitly identifies assumptions about dataset bias and leads directly to our theoretical results in Section 4. We believe that Proposition 4.8 is both the simplest and most general FRL impossibility result, and we credit this simplicity to our proposed causal setup. We think our setup and proof techniques will be useful for researchers aiming to prove other related results in the field.
>
> Third, our consideration of the distribution shift paradigm goes beyond much of the previous literature on fairness-accuracy tradeoffs, which often focuses on tradeoffs in the IID setting (notably [1,2] criticise the typical focus on the IID setting). While we did not include theoretical results for the distribution shift paradigm, our hypotheses and empirical results raise interesting new questions for the field and will help to spark future research directions.
>
> Tradeoffs of fair representation learning have indeed been studied previously, and we believe we give due credit to prior work in this space as thoroughly as possible. An important motivation for our paper was the amount of work that continues to use flawed methods and evaluation practices despite these previous theoretical results. We think that this is, in part, because the fairness literature is vast and complex, and many studies make different implicit assumptions, underscoring the need and timeliness of our work.
>
> [1] Wick M, Tristan JB. Unlocking fairness: a trade-off revisited. Advances in neural information processing systems. 2019;32.
>
> [2] Dutta S, Wei D, Yueksel H, Chen PY, Liu S, Varshney K. Is there a trade-off between fairness and accuracy? a perspective using mismatched hypothesis testing. International conference on machine learning 2020 Nov 21 (pp. 2803-2813). PMLR.

---

> > ### Author Response · Authors · 2024-11-20
> > **Response to Reviewer Comments (continued)**
> >
> > > "**This paper mainly talks about group fairness, what about counterfactual fairness? Is the counterfactual fairness also impossible in the iid setting?**"
> >
> > This is a very interesting point. In our paper, we focus on statistical notions of (group) fairness, as this is the explicit aim of FRL. These notions of fairness are indeed closely related to stronger causal notions of fairness such as counterfactual fairness [1]. However, the connections have some subtleties (and are a matter of some debate! [2,3]), so we felt that we would not be able to do this topic justice in the scope of our paper.
> >
> > That being said, it is true that counterfactual fairness implies demographic parity in many cases (but surprisingly, not all [3]). Under those circumstances, we would be confident to claim that Proposition 4.8 also applies to counterfactually fair predictors since the set of counterfactually fair predictors would be a subset of the FRL predictors that Proposition 4.8 applies to. We will add a brief discussion on the reviewer’s point, which will make an exciting direction for future work.
> >
> > [1] Anthis J, Veitch V. Causal context connects counterfactual fairness to robust prediction and group fairness. Advances in Neural Information Processing Systems. 2024 Feb 13;36.
> >
> > [2] Rosenblatt L, Witter RT. Counterfactual fairness is basically demographic parity. Proceedings of the AAAI Conference on Artificial Intelligence 2023 Jun 26 (Vol. 37, No. 12, pp. 14461-14469).
> >
> > [3] Silva R. Counterfactual Fairness Is Not Demographic Parity, and Other Observations. arXiv preprint arXiv:2402.02663. 2024 Feb 5.
> >
> > > "**This work shows that distribution shift is closely related to group fairness, and there are some results in the area of domain generalization / adaptation. Is there any method closely related to the group fairness? How does different types of distribution shift (class imbalance, covariate shift, concept shift ...) in the dataset related to the three types of disparities?**"
> >
> > We are strongly interested in both of these questions at the moment, and we think there is a rich opportunity to explore the connections between fairness, distribution shift, and domain generalisation.
> >
> > On one level, FRL and domain generalisation/adaptation techniques (especially methods based on adversarial training and disentanglement) share many similarities. Often, the problem setup in both of these fields is very similar, with fairness using a ‘sensitive attribute’ and domain adaptation using a ‘domain’ or ‘environment’ variable. This similarity gives us hope that our results may be insightful in fields beyond fairness, however, we do not consider such claims within the scope of this paper.
> >
> > Another key point to note is that in our formulation, there are two simultaneous shifts: a disparity across groups (e.g. prevalence, presentation, or annotation disparities, as enumerated in Fig. 1) and a potential shift across train/test domains (i.e. training on biased data and testing on unbiased data in the distribution shift paradigm). We can thus relate our problem to the traditional distribution shift setup in the following way by extending the framework of [1] for instance.
> >
> > To illustrate this, consider the joint distribution over training and testing datasets by using the binary indicator variable $T$ to distinguish between them. We can now decompose the shift across groups and domains like so:
> >
> > $I(X, Y, A; T) = I(X; T) + I(Y; T \mid X) + I(A; T \mid X,Y)$
> >
> > In this formulation, the LHS term represents the overall distribution shift, and the terms on the RHS represent covariate shift, label shift, and attribute shift, respectively. If the selection $T$ yields no information about the joint distribution then the training and test distributions are IID, i.e. $I(X, Y, A; T) = 0$. Different factorisations of this joint mutual information imply different data-generating processes and correspond to the various shifts shown in Fig. 1. Furthermore, different selection effects can be represented by the functional relation between $(X, Y, A)$ and the selection variable $T$. For instance, an attribute-based selection effect could be represented by the causal mechanism $T := f(A, U)$, where $U$ is an exogenous noise variable. There are various other combinations possible, including multivariate selection effects: $T := f(X, Y, A, U)$, or ones consisting only of exogenous noise: $T := f(U)$, which would represent the unbiased setting. Federici et al (2021) derive some practical upper bounds on the (latent) concept shift quantity which, under some reasonable assumptions, are guaranteed to minimise concept shift. In our view, deriving practical bounds for other types of distribution shifts of the sort studied in the present work and beyond constitutes fertile ground for future research.
> >
> > [1] Federici M, Tomioka R, Forré P. An information-theoretic approach to distribution shifts. Advances in Neural Information Processing Systems. 2021 Dec 6;34:17628-41.

---

> > > ### Comment · Reviewer_svct · 2024-11-22
> > >
> > > Thank you for the clarification, I have raised my score.

---

### Meta-Review · Area_Chair_Qnf8 · 2024-12-20

**Metareview:**

This paper makes a great and solid contribution toward understanding how fair representation learning fares for performance-sensitive tasks under different fair learning paradigms.  Specifically, three main fairness paradigms are considered in this paper, namely enforcing group parity, maximizing subgroup IID performance, and generalizing to unbiased distributions. The authors were able to characterize the limitations of existing methods. The key theoretical insight is that enforcing group fairness and achieving harm to all groups can not be satisfied simultaneously in the IID setting, and suggests that the success of fairness learning methods depends heavily on shifts in test data distribution. The above analysis leads to the hypothesis that performance is also influenced by the underlying causal structure and the amount of sensitive information available in the training dataset. The authors experimentally verified their findings.

**Additional Comments On Reviewer Discussion:**

Reviewers were unanimous about the paper's contributions

---

### Decision · Program_Chairs · 2025-01-22

Accept (Poster)